# Foulant Identification and Performance Evaluation of Antiscalants in Increasing the Recovery of a Reverse Osmosis System Treating Anaerobic Groundwater

**DOI:** 10.3390/membranes12030290

**Published:** 2022-03-02

**Authors:** Muhammad Nasir Mangal, Sergio G. Salinas-Rodriguez, Jos Dusseldorp, Bastiaan Blankert, Victor A. Yangali-Quintanilla, Antoine J. B. Kemperman, Jan C. Schippers, Walter G. J. van der Meer, Maria D. Kennedy

**Affiliations:** 1IHE Delft Institute for Water Education, Water Supply, Sanitation and Environmental Engineering Department, Westvest 7, 2611 AX Delft, The Netherlands; s.salinas@un-ihe.org (S.G.S.-R.); jancschippers@gmail.com (J.C.S.); m.kennedy@un-ihe.org (M.D.K.); 2Faculty of Science and Technology, University of Twente, 7500 AE Enschede, The Netherlands; a.j.b.kemperman@utwente.nl (A.J.B.K.); w.g.j.vandermeer@utwente.nl (W.G.J.v.d.M.); 3Oasen Drinkwater, Nieuwe Gouwe O.Z. 3, 2801 SB Gouda, The Netherlands; jos.dusseldorp@oasen.nl (J.D.); bastiaan.blankert@kaust.edu.sa (B.B.); 4Water Desalination and Reuse Center (WDRC), Biological and Environmental Science and Engineering Division (BESE), King Abdullah University of Science and Technology (KAUST), Thuwal 23955-6900, Saudi Arabia; 5Grundfos Holding A/S, Water Solutions, Poul Due Jensens Vej 7, 8850 Bjerringbro, Denmark; vyangali@grundfos.com; 6Faculty of Civil Engineering, Delft University of Technology, Stevinweg 1, 2628 CN Delft, The Netherlands

**Keywords:** calcium phosphate scaling, anaerobic groundwater, antiscalants, reverse osmosis, fouling

## Abstract

The objectives of this study are to assess the performance of antiscalants in increasing the recovery (≥85%) of a reverse osmosis (RO) plant treating anaerobic groundwater (GW) in Kamerik (the Netherlands), and to identify scalants/foulant that may limit RO recovery. Five different commercially available antiscalants were compared on the basis of their manufacturer-recommended dose. Their ability to increase the recovery from 80% to a target of 85% was evaluated in pilot-scale measurements with anaerobic GW and in once-through lab-scale RO tests with synthetic (artificial) feedwater. A membrane autopsy was performed on the tail element(s) with decreased permeability. X-ray photoelectron spectroscopy (XPS) analysis indicated that calcium phosphate was the primary scalant causing permeability decline at 85% recovery and limiting RO recovery. The addition of antiscalant had no positive effect on RO operation and scaling prevention, since at 85% recovery, permeability of the last stage decreased with all five antiscalants, while no decrease in permeability was observed without the addition of antiscalant at 80% recovery. In addition, in lab-scale RO tests executed with synthetic feed water containing identical calcium and phosphate concentrations as the anaerobic GW, calcium phosphate scaling occurred both with and without antiscalant at 85% recovery, while at 80% recovery without antiscalant, calcium phosphate did not precipitate in the RO element. In brief, calcium phosphate appeared to be the main scalant limiting RO recovery, and antiscalants were unable to prevent calcium phosphate scaling or to achieve a recovery of 85% or higher.

## 1. Introduction

One of the main developments in water treatment over the last few decades has been the advent of reverse osmosis (RO) technology. Due to the continuous development of RO, the decreasing prices of membrane technology, and its small footprint and excellent removal of contaminants (e.g., organic micro pollutants (OMPs), viruses, etc.) [1], the use of RO has been increasingly applied in the treatment of groundwater (GW) and surface water, which are the main sources for producing drinking water in many countries worldwide. For instance, in the Netherlands, over 60% of the produced drinking water by Dutch water supply companies is obtained from the treatment of GW [2,3], and several of these companies have adopted (or are investigating) the use of RO technology to produce high-quality drinking water.

Though RO technology has gained popularity in the water treatment sector and is widely accepted, it still faces some challenges, such as membrane fouling, that need to be addressed. Membrane fouling has an adverse effect on the operation of RO, including, but not limited to, the permeability loss of the membranes, increased pressure needs leading to higher operating expenses, an increase in salt passage of the RO permeate, and a shorter membrane lifetime as a result of frequent cleanings [4]. In RO processes, various types of fouling can be encountered, such as particulate fouling, organic fouling, biofouling, and scaling [5,6,7,8]. Particulate fouling is caused by the deposition of colloidal and suspended material (silt, clay, iron oxides, etc.) present in the RO feed onto the membrane surface [9,10,11]. Organic fouling is usually encountered when the RO feed contains high concentrations of natural organic matter (e.g., humic substances (HS)) [12,13]. Biofouling is the attachment and growth of microorganisms on the feed spacer and membrane surface in RO processes [14,15,16]. In GW applications, especially the RO treatment of anaerobic GW, biofouling is not encountered [1,17]. Lastly, scaling refers to the crystallization and precipitation of sparingly soluble salts on the membrane surface that can occur when the concentration of the salts on the membrane surface exceeds their solubility limits [4].

Scaling is a major challenge in brackish water RO applications (BWRO), and is typically the key barrier in operating RO systems at high recovery rates [18]. Maximizing recovery in BWRO is highly desirable to minimize the total electrical consumption per unit volume of permeate, maximize water production, decrease the amount of concentrate, and lower the use of pretreatment chemicals and their related costs [19]. At high recoveries, the concentration of the dissolved inorganic compounds in the concentrate can increase considerably, as much as four to ten times for recoveries in the 75–90% range, and, consequently, exceed the solubility limits for several types of salts, which can lead to membrane scaling [20].

Depending on the inorganic ion composition of the RO feed, various compounds such as calcium carbonate, calcium phosphate, silica, etc. can precipitate in RO installations. Calcium carbonate is one of the most commonly encountered scales on RO membranes. The formation and degree of calcium carbonate scaling depend on the calcium and bicarbonate concentrations [20,21], as well as on pH and temperature [22]. When the RO feed contains a high concentration of calcium and orthophosphate ions, calcium phosphate scale can form on the membrane surface [23,24]. Calcium phosphate can be mainly encountered in water reuse applications, as well as in the RO treatment of GW. Calcium phosphate can exist in an amorphous form and in various crystalline forms [25]. In RO applications, the amorphous phase of calcium phosphate is reported to be responsible for flux decline [23,26]. Silica degrades membrane performance by precipitating as colloidal silica or as metal silicates with ions such as calcium, magnesium, aluminium, etc. [20,27,28]. In GWRO applications, aluminium silicates (e.g., clay in colloidal form) are one of the most commonly encountered compounds which could be present in the RO feedwater and can also cause permeability decline in the first stage [29].

Adding antiscalant to the RO feed is one of the most effective and widely used methods for preventing scaling and achieving high recoveries in RO applications. [30,31,32]. One factor that makes antiscalant addition appealing is the low dose required to overcome scaling [20]. Antiscalants disrupt the crystallization process; more specifically, they hinder the nucleation phase and/or retard the growth phase of crystallization [20,22,33], allowing for higher recovery in RO applications [34]. There are a variety of commercial antiscalants available that are designed to combat specific types of scale, and the most common ones used in RO applications are phosphonates, polycarboxylates, and biobased antiscalants [20,35]. The selection of antiscalants in RO applications depends on the feed water composition. The antiscalant dose is generally recommended by the antiscalant suppliers, which they calculate using their projection programs. The projection programs of the antiscalant suppliers also predict the maximum achievable recovery and identify the scalants that may limit RO recovery.

This study was performed in the context of the realization of a future RO plant by a Dutch water supply company (Oasen Drinkwater) in treating anaerobic groundwater for drinking water production. One of the main differences between anaerobic groundwater and aerobic groundwater is the presence of high concentrations of iron in its soluble (ferrous) form due to the absence of oxygen. On the other hand, a high concentration of iron in aerobic groundwater is not expected, since in the presence of oxygen, the formed iron (III) oxide particles (from the oxidation of ferrous to ferric) are retained in the soil pores.

Due to the anticipated increase in salinity and higher standards in the removal of OMPs, the water company aims to replace the existing conventional plant (spray aeration on the surface of rapid sand filters, tower aeration, pellet softening, rapid sand filtration, and granular activated carbon filtration) with RO. The abstraction of anaerobic groundwater and the discharge of concentrate are limited by strict regulations. It is preferable that the water loss, i.e., the concentrate waste stream, in the new RO is less (or at least equal) to that in the current conventional treatment plant. The current conventional treatment plant has a water loss of about 15%. It is therefore desirable, in this situation, to operate the RO plant at 85% recovery (or higher), which corresponds to the water loss of the current conventional treatment plant. Increasing RO recovery will result in lower groundwater abstraction and discharge of concentrate water for a given permeate water production, and thus less water loss (waste). Furthermore, increasing recovery will reduce the specific energy consumption and costs associated with the disposal of concentrate water.

The objectives of this work are:

(a)To identify the foulant/scalants that would precipitate in the RO unit at 85% recovery.(b)To examine the role of antiscalants in increasing the RO recovery to at least 85%.

In this study, an RO pilot unit was operated with and without antiscalant at 80–85% recoveries. Membrane autopsy was carried out to identify the scalants/foulant. To explore the effectiveness of the antiscalants, we combined results from the RO pilot, operating with anaerobic GW, and a once-though lab-scale RO system operating with synthetic feedwater.

## 2. Materials and Methods

### 2.1. Feedwater (Anaerobic GW) Composition

The RO feed was anaerobic GW which was obtained from several wells of a (conventional) water treatment plant in Kamerik, the Netherlands. Table 1 shows the composition of the feedwater, which contains high concentrations of ferrous iron. It should be noted that ferrous iron by itself will not cause membrane fouling because it is very soluble. However, if oxygen enters the feedwater, ferrous will oxidize to iron (III), or ferric, and form particle deposits on the membrane surface and spacers, resulting in a decrease in permeability and a rise in pressure drop. Therefore, maintaining the anaerobic status of the feedwater is essential.

The water analysis was carried out by a commercial lab (Vitens Laboratorium, the Netherlands). The major fraction (approximately 62%) of the dissolved organic carbon (DOC) in the GW was humic substances (HS) of the fulvic acid (FA) type, which was identified with liquid chromatography–organic carbon detection (LC–OCD) (DOC-Labor, Germany).

The RO feedwater data (Table 1) was entered into the projection programs of seven different antiscalant manufacturers (names are not included in this paper) to identify the suppliers’ recommended maximum achievable recovery and the scaling compound(s) which may limit RO recovery. The projection programs were also used to assess the scaling potential of the RO concentrates at various recoveries (with and without antiscalant addition).

#### Tested Antiscalants to Increase RO Recovery to at Least 85%

Based on the RO feed analysis (Table 1), various antiscalants were recommended by the antiscalant manufacturers with which the RO recovery could be increased to 85%. Table 2 lists the arbitrary names of the tested antiscalants, along with some basic information provided by the antiscalant suppliers.

### 2.2. RO Pilot

Figure 1 depicts a schematic representation of the RO pilot plant. The RO installation had three stages where the pressure vessel configuration for each stage could be varied. Each pressure vessel contained three hydranautics membrane elements (ESPA2-LD-4040). The anaerobic GW (feedwater) was passed through a cartridge filter (10 µm) and then fed to the RO unit. The RO installation was operated at constant permeate water production. To assess the occurrence of scaling, the average normalized (temperature corrected) permeability (K_w,_
Appendix A) of the last stage was monitored.

In the first set of experiments, the RO pilot was operated without antiscalant, as described in Table 3, to realize which compounds will precipitate in the RO unit in the absence of antiscalant. After observing a permeability drop (> 20%) in the last stage, the tail membrane element was taken for autopsy. In the second set of experiments (as shown in Table 3), the RO unit was operated at 85% recovery with various antiscalants. These experiments were conducted to recognize if a recovery of 85% (or higher) could be achieved with the use of antiscalants for the RO unit in Kamerik, since according to some antiscalant suppliers, 85% recovery was an achievable recovery with their antiscalants. The average flux of the last stage in all experiments was in the 10–20 L/h/m^2^/bar range.

### 2.3. Foulant Characterization

After operating the RO unit without antiscalant, all three stages were flushed with RO permeate. As the membranes were in contact with anaerobic concentrate containing high concentrations of ferrous, flushing with RO permeate was necessary to avoid iron oxidation (and precipitation) while taking out the membranes for autopsy. Membrane autopsy was performed on the tail element of the third stage and first stage. It is worth mentioning that the membrane elements in the third stage were brand new, whereas the membrane elements in the first stage had been in use for over 5 years.

To identify the foulant/scalant which was responsible for the permeability decline of the RO, various techniques were employed, such as scanning electron microscopy with energy dispersive X-ray (SEM-EDX) spectroscopy (JEOL, JSM-6010LA), X-ray powder diffraction (XRD) (Bruker D8 Advance), fluorescence excitation–emission matrix (FEEM) spectrophotometry (FluoroMax-3), and X-ray photoelectron spectroscopy (XPS) (Quantera SXM-Scanning XPS microprobe).

SEM-EDX was used to visualize the foulant and to identify the mass percentages of the elements present in the foulant. To investigate whether the foulant disappears in acidic or basic solutions (or both), membrane coupons of the fouled RO with a total area of approximately 1000 cm^2^ were stirred for about 24 h at 35 °C in beakers containing either 0.05 M HCl or 0.05 M NaOH. Afterwards, the membrane coupons were flushed with demineralized water, dried, and then analyzed with SEM-EDX. In the case where foulant was not observed on the membrane coupons (after cleaning) in SEM-EDX analysis, the cleaning solutions then were filtered through 0.45 µm (cellulose acetate) filters. Afterwards, the 0.45 µm filters were flushed with demineralized water, dried, and analyzed with SEM-EDX to find out if the foulant dissolved in the cleaning solutions or was just physically detached from the membrane.

XRD analysis was performed on the fouled RO membrane to examine if the foulant was crystalline, and, if so, which scales were present on the fouled membrane. FEEM analysis was used to examine the cleaning solutions (0.05 M HCl and 0.05 M NaOH) to identify the presence of HS on the fouled RO membrane. XPS was used to obtain the binding energies of the foulant present on the membrane surface and to identify the foulant.

### 2.4. Lab-Scale RO Unit

Calcium phosphate was expected to be one of the scalants limiting RO recovery in Kamerik (explained later in the Section 3). Due to the complexity of the water composition of the anaerobic groundwater, i.e., presence of iron and HS, it was necessary to execute once-through lab-scale RO experiments (Figure 2) with synthetic solutions (in the absence of iron and HS) to assess the ability of antiscalants in hindering calcium phosphate scaling. As presented in Table 4, the lab-scale RO experiments were performed with the synthetic concentrate solutions of 80 and 85% recovery in the absence and presence of antiscalants. The synthetic RO concentrate solutions had similar pH, calcium, and phosphate concentrations to the real RO concentrates in Kamerik.

The synthetic concentrate solutions were prepared by dosing Ca^2+^, HCO_3_^−^, PO_4_^3−^, and NaOH from their stock solutions to the demineralized water (demi-water). The synthetic concentrate solutions were stirred in a reactor at 200 rpm for less than 1 min before being fed to the membrane element. The volume of the synthetic concentrate in the reactor was kept constant at 1 L, and the flows entering and exiting the reactor were both kept constant at 90 L/h, allowing for a residence time of less than 1 min.

An OSMO inspector unit (Convergence Industry B.V., the Netherlands) was used to feed the synthetic concentrate solution to a TW30-1812-50 RO element (OsmoPure Water Systems). An Atrato ultrasonic flow meter was installed in the OSMO unit, which could measure permeate flow rates ranging from 0.12 L/h to 30 L/h. A new TW30-1812-50 RO element was used in each experiment.

In all tests, the initial recovery of the RO element was 5–6% and the permeate flux was 13–15 L/m^2^/h. The cross-flow velocity was between 10 and 12 cm/s. All lab-scale RO tests were executed at room temperature (20–22 °C).

## 3. Results and Discussion

### 3.1. Maximum Achievable Recovery Based on Antiscalant Suppliers’ Projection Programs

In this section, the maximum achievable recovery and the scalants that limit RO recovery according to the projection programs of the antiscalant suppliers are discussed. In addition, the scaling potential (of some commonly encountered scalants) at 80 and 85% recovery with and without antiscalant (according to the projection software) is presented.

Figure 3a shows the maximum achievable recoveries in the presence of antiscalants for the RO unit, which were determined by the projection programs of seven different antiscalant suppliers. As can be seen, the recommended maximum achievable RO recovery was different for all the projection programs of the antiscalant suppliers. In addition, the scalant compound, which may limit RO recovery, was not the same, according to the projection programs of different suppliers. For instance, according to the projection programs of suppliers A, E, C, and G, recovery of the RO unit was limited due to calcium carbonate scaling, while calcium phosphate scaling was limiting RO recovery according to suppliers B, D, and F. The maximum achievable RO recovery (limited due to calcium carbonate) was 89% according to suppliers E and G, and 83% and 87% according to suppliers A and C, respectively. The maximum achievable RO recovery (limited due to calcium phosphate) was 77%, 80%, and 85% according to suppliers B, D, and F, respectively. From Figure 2a, one can clearly recognise that the projection programs of the antiscalant suppliers are not consistent in identifying the maximum RO recovery.

Figure 3b,c present the scaling tendency (performed with the projection program of supplier B) of some commonly encountered scalants in the RO concentrate at 80 and 85% recovery, respectively. The scaling tendency of the RO concentrates at 80 and 85% recoveries is presented because the RO pilot, in this study, was operated initially at 80% recovery and then at 85% recovery. The scaling tendency is shown both with and without the addition of antiscalant to the RO feed.

The projection program of supplier B advised that at 80% recovery without antiscalant, the RO unit may experience calcium carbonate, barium sulphate, and calcium phosphate scaling, while with the addition of antiscalant to the RO feed, none of the mentioned scalants will precipitate in the RO unit. Furthermore, at 85% recovery with no antiscalant, the program indicated that silica may also precipitate together with calcium carbonate, barium sulphate, and calcium phosphate. The program suggested that, with the addition of antiscalant, calcium carbonate, barium sulphate, and silica may not precipitate in the RO unit at 85% recovery, while calcium phosphate may precipitate even with the addition of antiscalant.

In short, with the use of projection programs, it is not clear if the RO unit in Kamerik can be operated at 85% recovery (or higher), as the highest achievable recovery according to antiscalant suppliers varied from 77 to 89%.

### 3.2. Foulant (Scalant) Characterization

In this section, the aim was to understand which compounds may limit RO recovery. For this, the RO unit was initially operated at various recoveries without antiscalant, and in the case of a decrease in permeability, the membranes were taken out for autopsy to identify the scalants responsible for the observed permeability decline in the absence of antiscalants. After identifying the scalant(s), the RO was operated with various antiscalants of different suppliers in an attempt to inhibit the precipitation of those scalants and to maximize RO recovery (presented later in Section 3.3).

#### 3.2.1. RO Operation at 80–85% Recoveries in the Absence of Antiscalants

Figure 4 presents the average normalized permeability of the first, second, and third stages of the RO unit at 80–85% recoveries when no antiscalant was added to the RO feed.

At 80% recovery without antiscalant, the normalized permeability remained constant, where calcium carbonate, barium sulphate, and calcium phosphate had the tendency to scale the RO unit, according to the projection program of supplier B. On the other hand, at 85% recovery, the normalized permeability of the last stage decreased in the absence of antiscalants. At 85% recovery, the normalized permeability of the second stage remained constant, where the total recovery of the first and second stages was approximately 77%. As no decrease in membrane permeability of the last stage at 80% recovery without antiscalant was observed, it was expected that permeability would remain constant at 77% recovery. Surprisingly, the average normalized permeability of the first stage with a recovery of 52% had a slight decreasing trend (with a—0.005 slope). The slight decrease in the first stage may not be due to the deposition/precipitation of newly formed particles/crystals. It could be that the RO feed contained particles that were deposited in the first stage and caused permeability decline. To understand what compounds caused permeability decline in the third stage and in the first stage, the tail elements of the mentioned stages were taken out for autopsy.

#### 3.2.2. Membrane Autopsy

##### SEM-EDX of the Tail Element of the First Stage

The SEM and EDX analyses of the tail element of the first stage (with decreased permeability) are presented in Figure 5a,b, respectively. The SEM picture showed that the membrane surface was covered by deposits. The EDX results in Figure 5b indicated that the deposited material on the membrane surface consisted of aluminium, silicon, and iron. In the EDX analysis (Figure 5b), only those elements that are not part of the membrane material are presented. The presence of aluminium and silicon on the membrane surface could be attributed to clay particles that might be present in the RO feedwater that were not retained by the 10 µm cartridge filter. The presence of iron may be related to the deposited clay particles, as iron may be present in the composition of the clay particles, and/or to the iron oxide particles that may be present in very low concentrations in the RO feed.

Figure 5c,d present the SEM image and the EDX of the membrane cleaned with 0.05 M NaOH solution, respectively. It was observed that the NaOH solution was unable to remove the deposited particles from the membrane surface, and the layer (after cleaning) still consisted of aluminium, silicon, and iron. On the other hand, the deposited particles disappeared when the membrane was stirred in 0.05 M HCl solution (Figure 5e). In the EDX analysis (Figure 5f), no aluminium, silicon, and iron were detected. This showed that the deposited layer was removed with HCl solution. At this point, it was not clear if the deposited particles were dissolved in the HCl solution and/or were detached from the membrane surface due to mechanical forces.

In Figure 6, the SEM-EDX analysis of the retained deposits on the 0.45 µm filter (after filtering the 0.05 M HCl cleaning solution) is shown. As can be seen, the retained deposits consisted of aluminium, silicon, and iron suggesting that clay particles from the membrane surface were not dissolved in the HCl solution, but actually were detached from the membrane surface. Figure 6b also indicated that a part of the iron on the membrane surface of the tail element of the first stage (Figure 5a,b) could be linked to clay particles, since if all iron was present as iron oxides, then it should have been dissolved in 0.05 M HCl solution. In Figure 6b, the mass percentage of iron (on the filter surface) was approximately half the mass percentage of silicon. However, in Figure 5b, the mass percentage of iron on the membrane surface of the tail element of the first stage was higher than silicon. This may indicate that the additional mass percentage of iron in Figure 5b could be due to the iron oxide particles.

In brief, based on the SEM-EDX results of Figure 5 and Figure 6, it may be concluded that the deposition of clay and iron oxide particles (present in the RO feed) were contributing to the slight permeability decline of the first stage.

##### SEM-EDX of the Tail Element of the Third Stage

In Figure 7a,b, SEM-EDX analysis of the fouled/scaled RO membrane of the tail element of the third stage is shown. As can be seen, the membrane surface apparently was covered with an amorphous layer. According to the EDX analysis, the foulant was composed of calcium, phosphorous, and iron. A small mass percentage of manganese could also be observed. In the EDX analysis, aluminium and silicon were not observed on the membrane surface, which suggested that clay particles did not contribute to the permeability decline of the third stage at 85% recovery. According to the antiscalant projection program of supplier B (Figure 3c), calcium carbonate, barium sulphate, calcium phosphate, and silica have the potential to scale the RO at 85% recovery in the absence of antiscalant. As silica was not observed on the membrane surface in the EDX analysis (Figure 7b), it can be concluded that silica scaling did not occur in the RO unit at 85% recovery without antiscalant.

Furthermore, barium sulphate scaling did not occur at 85% recovery, as barium was not present on the fouled membrane surface (Figure 7b). Based on a study by Boerlage, et al. [37] where they reported that barium sulphate has very slow precipitation kinetics and its precipitation is hindered by humic substances (HS), it was expected that the barium sulphate scale would not occur in the RO unit in Kamerik. In the anaerobic groundwater of Kamerik (RO feed), the concentration of HS was approximately 5.3 mg/L (Table 1), which might inhibit barium sulphate scaling in the RO unit. Several researchers [38,39,40,41,42] have reported that HS significantly hinder the formation of calcium carbonate. Also, in a study by the current authors [36], it was demonstrated that HS, as well as the phosphate present in the GW, were able to inhibit calcium carbonate scaling in the RO unit in Kamerik. We showed that due to the presence of HS and phosphate, the induction time of the real RO concentrate at 80% recovery was longer than 7 days, whereas at the same supersaturation level in the absence of phosphate and HS, the induction time of the synthetic RO concentrate of 80% recovery was around 1 h. Due to the long induction time (>168 h) of the RO concentrate of 85% recovery in the absence of antiscalants (Appendix A), it is also expected that calcium carbonate scaling would not occur at 85% recovery when antiscalants are not added to the RO feed. In the XRD analysis (Appendix A), calcium carbonate crystals were not detected, which also indicates that calcium carbonate scaling was not the reason for the permeability decline in the last stage, as seen in Figure 4c.

In Figure 7b, the mass percentage of calcium and phosphorous on the membrane surface could be attributed to (amorphous) calcium phosphate scaling, which was also predicted by the projection programs of suppliers B, D, and F (Figure 3a). However, the presence of iron on the fouled membrane is not clear. As aluminium and silicon were not observed on the fouled membrane of the third stage (Figure 7a), the presence of iron could not be due to clay particles. One may suggest that the presence of iron on the membrane could be due to the presence of iron oxide particles in the RO feed. The presence of iron (and also of calcium and phosphorous) on the fouled membrane is elucidated via XPS analysis later.

Figure 7c,d presents the SEM-EDX analysis of the fouled membrane which was cleaned with 0.05 M NaOH solution. As can be seen, the alkaline solution (0.05 M NaOH) was unable to remove the foulant from the membrane surface. After cleaning at high pH, calcium, phosphorous, and iron were still present on the membrane surface. On the other hand, the acidic solution (0.05 M HCl) was able to remove the foulant from the membrane surface, as can be seen from the SEM image (Figure 7e) and the EDX analysis (Figure 7f). As calcium, phosphorous, and iron were not detected on the membrane surface (Figure 7e), the acidic solution was filtered through a 0.45 µm filter and then the filter was examined by SEM-EDX. It was found that the foulant (composed of calcium, phosphorous, and iron) on the tail element of the third stage was dissolved in the acidic solution (Appendix A).

In brief, from Figure 7, one can conclude that the foulant (on the membrane surface of the tail element of the third stage) was mainly inorganic, which could be dissolved in an acidic solution (0.05 M HCl), but not in an alkaline solution (0.05 M NaOH).

##### XPS Analysis of the Tail Element of the Third Stage

The fouled membrane (tail element of the third stage) was analysed with XPS analysis in an attempt to identify the scalant(s) to which calcium, phosphorous, and iron could be attributed. The survey spectrum of the analysis is shown in Appendix A. Calcium, phosphorous, iron, manganese, oxygen, nitrogen, sulphur, and carbon were all detected. In Table 5, the average atomic concentrations (obtained with the XPS analysis) from four different spots of the fouled membranes are shown. As the foulant layer was thin, nitrogen, sulphur, carbon, and partly oxygen could be attributed to the polyamide membrane. In the survey spectrum, aluminium and silicon were not observed, which further verified the results of SEM-EDX of Figure 7a,b, showing that clay particles were not (mainly) present on the membrane surface.

In Table 6, the binding energies of calcium, phosphorous, iron, and manganese (present on the fouled membrane surface) are given. In addition, for the determined binding energies, the identified compounds based on the XPS database of the National Institute of Standards and Technology (NIST) [43] are included in Table 6. The binding energies of calcium and phosphorous were 347.2 eV and 132.9 eV, respectively, which corresponded to calcium phosphate compounds in the NIST database. This suggested that the presence of calcium and phosphorous on the fouled membrane surface (tail element of the third stage) can be attributed to calcium phosphate scaling, as predicted by the projection programs of suppliers B, D, and F (Figure 3a).

In the XPS analysis, the binding energies of iron and manganese were found to be 711.1 eV and 624.4 eV, respectively. According to the NIST database, the presence of iron and manganese on the fouled membrane of the tail element of the third stage could be due to the precipitation of oxidized iron and manganese. This may suggest that iron oxide particles were present in the raw water (RO feed) before entering the plant. It may also be that iron oxide particles were formed (and precipitated on the membrane surface) when the anaerobic RO concentrate (containing a ferrous concentration of approximately 57 mg/L) came into contact with the aerobic RO permeate during flushing of the last stage, which was needed for membrane autopsy. If iron oxide particles were formed during flushing, then they were not responsible for the observed permeability decline of the last stage in Figure 4c.

Nevertheless, from the SEM-EDX analysis (Figure 7a,b) and XPS analysis (Table 5 and Table 6), it can be concluded that calcium phosphate scaling was one of the reasons for the permeability decline of the third stage at 85% recovery (Figure 4c) when no antiscalant was added to the RO feed.

##### FEEM Analysis of the 0.05 M HCl and 0.05 M NaOH Cleaning Solutions

As discussed earlier in Section 2.1 (Table 1), the dissolved organic carbon (DOC) concentration in the RO feed was approximately 8.6 mg/L, of which 5.3 mg/L were HS. This means that the concentration of HS in the RO concentrate at 85% recovery could increase to approximately 35 mg/L. In RO processes, HS are recognized by various researchers [44,45,46,47] to cause membrane fouling. Therefore, it was essential to investigate if HS also contributed to the permeability decline of the third stage (Figure 4c).

In Figure 8, fluorescence excitation-emission matrix (FEEM) analysis of the RO concentrate (at 85% recovery) and the HCl and NaOH cleaning solutions of the tail elements of the third stage are shown. In this study, the locations of the EEM peaks of humic acid (HA) and fulvic acid (FA) were based on the EEM regions reported by Chen et al. [48]. The HA peaks were observed in the 380–500 and 250–400 nm emission and excitation wavelength ranges, respectively, while the 380–400 and 200–250 nm emission and excitation wavelength ranges, respectively, were attributed to FA [48].

As illustrated in Figure 8a, peaks for both HA and FA (with high intensity) were observed in the RO concentrate, which verifies the presence of HS in the anaerobic GW. However, HA and FA peaks were not visible in the cleaning solutions of the tail element of the third stage, as shown in Figure 8b (NaOH solution) and Figure 8c (HCl solution). The absence of HA and FA peaks in the HCl and NaOH solutions suggests that HS were likely not responsible for the permeability decline of the third stage of the RO unit in Figure 4c.

### 3.3. Role of Antiscalants in Increasing RO Recovery to 85%

In the previous section, it was found that calcium phosphate was one of the scalants leading to the permeability decline of the RO unit at 85% recovery and was limiting the RO recovery when no antiscalant was added to the RO feed. The section aimed to investigate if the permeability decline at 85% could be prevented with the addition of antiscalants.

#### 3.3.1. RO Pilot Operation at 85% Recovery with Various Antiscalants

In this section, five different antiscalants (Table 2) were tested that were claimed by the antiscalant suppliers to have excellent performance in preventing calcium phosphate scaling at 85% recovery. The aim was to increase the RO recovery to 85% (and even higher) with the use of antiscalants. Therefore, the RO pilot was operated at 85% recovery (Table 3) with the suppliers’ recommended antiscalants and antiscalant doses.

Figure 9 shows the average normalized permeability of the third stage of the RO unit at 85% recovery in the absence and presence of the five tested antiscalants. As can be seen, none of the antiscalants could prevent the permeability decline of the third stage. After operating the RO unit with antiscalant (AS–5), the tail element of the third stage was examined with SEM-EDX, and the results were similar to those shown in Figure 7. It is worth mentioning that the permeability of the third stage also decreased at 83% recovery in the presence of antiscalants (Appendix A). The RO operation at 81 and 82% recovery was not investigated.

Some antiscalant suppliers claim that the effectiveness of their antiscalants is reduced when iron (II) is present in the RO feed. Furthermore, as the RO feed contained clay particles and probably iron oxide particles as well, it may be too early to conclude (based on the results of Figure 9) that antiscalants were not effective in preventing calcium phosphate scaling. It is worth mentioning that the tested antiscalants were claimed by the suppliers to have good performance in dispersing clay and iron oxide particles and not allowing them to deposit on the membrane.

Nevertheless, to understand if the tested antiscalants can prevent calcium phosphate scaling in the absence of iron (II), once-through lab-scale RO tests (Section 3.3.2) were performed with the synthetic concentrates of 85% recovery, where calcium phosphate was the only scalant that could cause permeability decline of the RO.

#### 3.3.2. Lab-Scale RO Tests with the Synthetic Concentrate of 85% Recovery with Various Antiscalants

In the previous section (Section 3.3.1), due to the complexity of the water chemistry of the anaerobic GW, i.e., presence of iron and HS, it was not possible to conclude if antiscalants were effective in preventing calcium phosphate scaling at 85% recovery. The aim of this section was to see if antiscalants could prevent calcium phosphate scaling in once-through lab-scale RO tests using synthetic RO concentrates (Table 4) that had similar pH, calcium, and phosphate concentrations to the real RO concentrates in Kamerik.

Figure 10a shows the normalized permeability of the TW30-1812-50 membrane element when fed with the synthetic concentrates of 80 and 85% recoveries in the absence of antiscalants (see Table 4 for the testing conditions). As can be seen, the normalized permeability of the membrane remained constant when fed with the synthetic concentrate of 80% recovery. This result is consistent with the RO pilot results (Figure 4c), where the normalized permeability in the third stage remained constant at 80% recovery without antiscalant. On the other hand, when the membrane element was fed with the synthetic concentrate of 85% recovery, the normalized permeability decreased sharply, as illustrated in Figure 10a. This result also verifies that calcium phosphate was one of the scalants that caused permeability decline of the third stage of the RO pilot in Figure 4c. A SEM image and an EDX analysis of the membrane with decreased permeability (Figure 10a) are given in Figure 10c,d, respectively. As can be seen, the membrane surface was covered with an amorphous layer (verified with XRD analysis (Appendix A)) of calcium phosphate particles. In the EDX analysis, the mass percentage of fluoride, aluminium, silicon, manganese, iron, and barium was zero, which was because they were not present in the synthetic concentrate of 85% recovery.

Figure 10b shows the normalized permeability of the RO when fed with the synthetic concentrate of 85% recovery in the presence of 33 mg/L (equivalent to 5 mg/L dose in the RO feed) of various antiscalants. The normalized permeability decreased by at least 10% with each antiscalant in a 3 h experimental period, indicating that none of the antiscalants could prevent calcium phosphate scaling. From Figure 10b, one can also observe that the poor performance of the antiscalants in preventing the permeability decline of the third stage of the RO unit in Figure 9 was not (mainly) due to the presence of a high concentration of iron (II). The permeability of the small RO element in Figure 10b decreased more sharply than the permeability of the third stage of the RO unit in Figure 9. One possible explanation is that the supersaturation level of calcium phosphate in the synthetic RO concentrate was slightly higher than that of the real RO concentrate because the once-through lab-scale RO experiments were performed at room temperature (approximately 22 °C), whereas the RO concentrate was at around 12 °C. Furthermore, Figure 9 shows the average permeability of three membrane elements, implying that the actual permeability decline of the last element of the RO unit could be sharper than the ones of third stage shown in Figure 9.

To summarize, calcium phosphate appeared to be one of the primary scalants precipitating in the RO unit in Kamerik, and the available antiscalants for calcium phosphate were unable to prevent calcium phosphate scaling and increase RO recovery to 85%.

## 4. Conclusions

In this study, the role of antiscalants in increasing the recovery of a future RO system of a Dutch water supply company, which will treat anaerobic groundwater in Kamerik (the Netherlands) for drinking water production, to at least 85% and the scalants that could limit RO recovery were investigated.

The following are the main findings of this study:The maximum achievable recovery and the scalant limiting the RO recovery varied according to the projection programs of the different antiscalant suppliers, with some pointing to calcium carbonate and others to calcium phosphate as the limiting scaling compound. The maximum achievable recovery according to antiscalant suppliers was ranging between 77% and 89%.Operation of the RO at 80–85% recoveries without antiscalant:The normalized permeability of the third stage remained constant during a 1 month experimental period when the RO pilot was operated at 80% recovery without antiscalant, whereas the normalized permeability of the third stage decreased when the RO pilot was operated at 85% recovery without antiscalant.Membrane autopsy of the tail element of the third stage:Calcium phosphate was the main scalant causing permeability decline at 85% recovery and limiting the RO recovery.Calcium carbonate was not responsible for the permeability decline of the third stage at 85% recovery.Role of antiscalants in increasing the RO recovery to 85% (and higher):In the RO pilot measurements, the tested antiscalants were found to be ineffective in increasing the RO recovery to 85% as the permeability of the third stage decreased with each of the tested antiscalants.In once-through lab-scale RO tests, none of the tested antiscalants could prevent calcium phosphate scaling when the RO element was fed with the synthetic concentrate of 85% recovery.

## Figures and Tables

**Figure 1 membranes-12-00290-f001:**
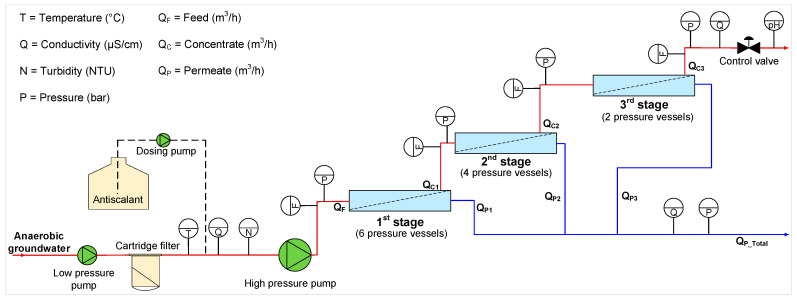
Schematic representation of the RO pilot installation in Kamerik [36].

**Figure 2 membranes-12-00290-f002:**
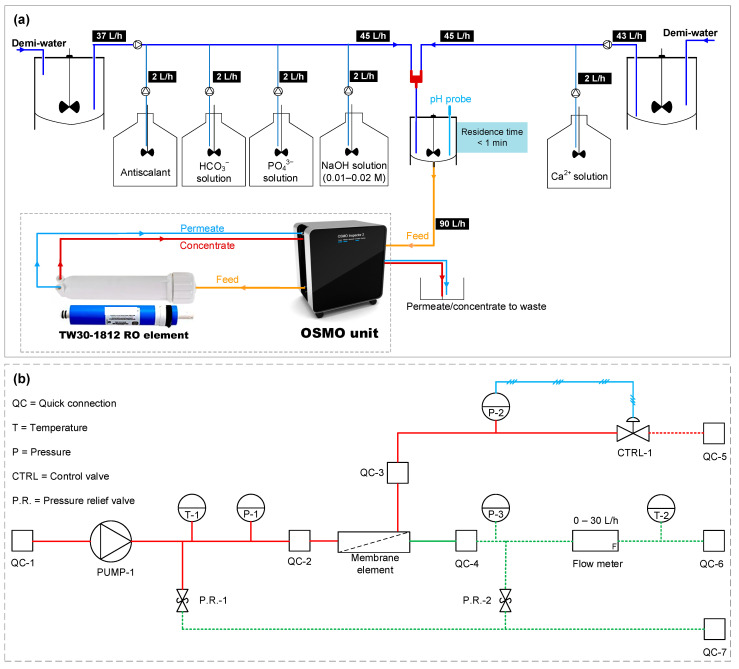
(**a**) Once-through lab-scale RO setup for testing the performance of antiscalants. (**b**) Piping and instrumentation diagram of the OSMO unit with the RO element [26].

**Figure 3 membranes-12-00290-f003:**
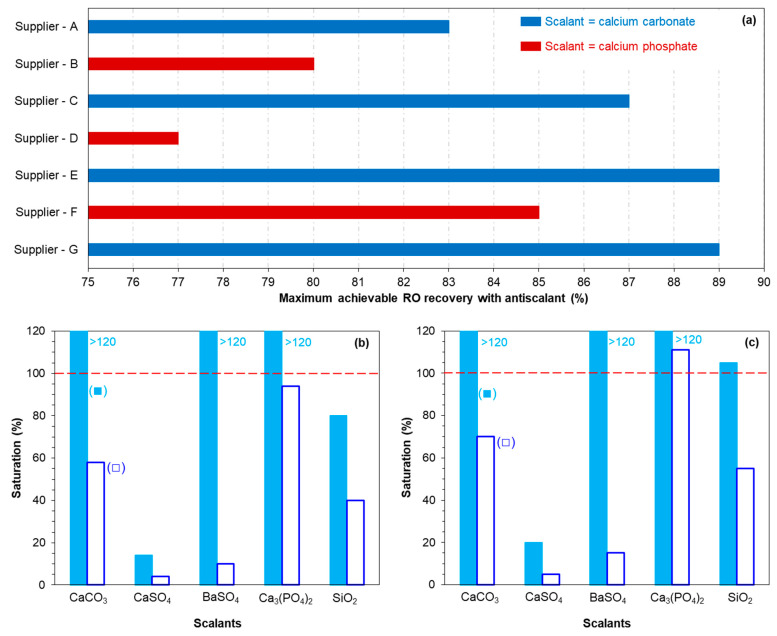
(**a**) Maximum achievable recovery of the RO unit in the presence of antiscalants determined with the projection programs of various antiscalant suppliers. (**b**) Scaling potential of commonly encountered scalants at 80% recovery of the RO unit (■) in the absence of antiscalant, and (□) in the presence of antiscalant. (**c**) Scaling potential of commonly encountered scalants at 85% recovery of the RO unit (■) in the absence of antiscalant, and (□) in the presence of antiscalant. (**b**,**c**) are determined with the projection program of supplier B.

**Figure 4 membranes-12-00290-f004:**
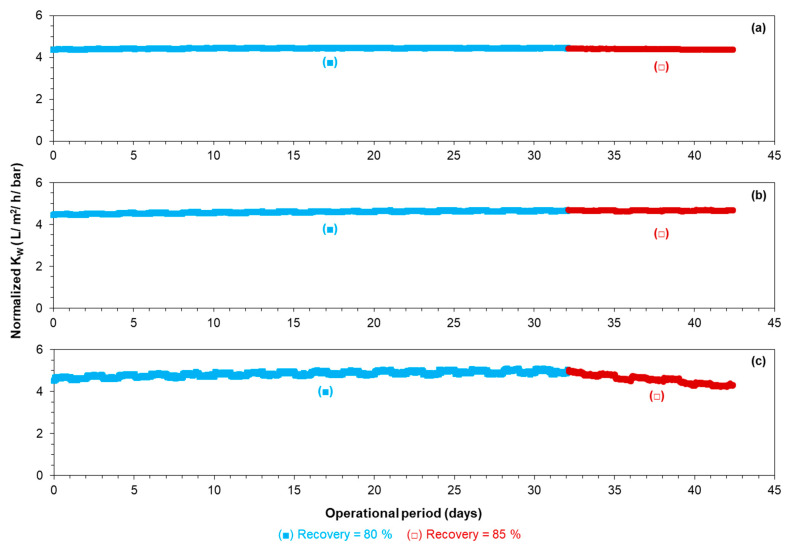
Average normalized permeability of the (**a**) first stage, (**b**) second stage, and (**c**) third stage of the RO unit at 80 and 85% recoveries without antiscalant addition.

**Figure 5 membranes-12-00290-f005:**
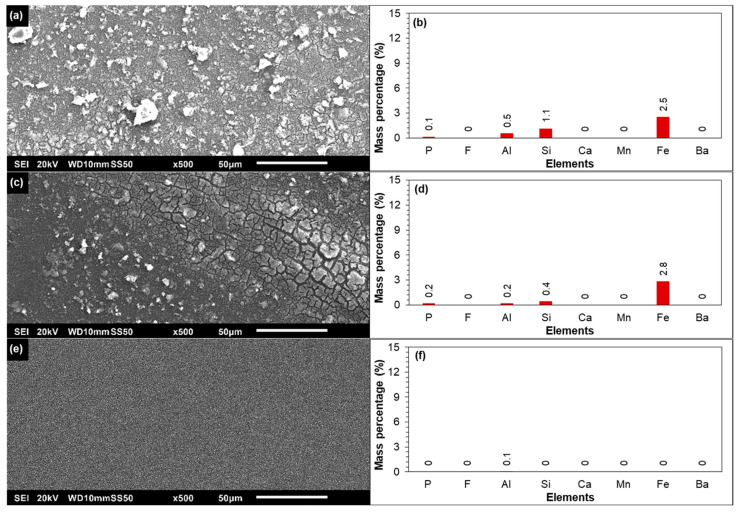
(**a**) SEM and (**b**) EDX analyses of the tail element of the first stage with decreased permeability. (**c**) SEM and (**d**) EDX analyses of the tail element of the first stage cleaned with a 0.05 M NaOH solution. (**e**) SEM and (**f**) EDX analyses of the tail element of the first stage cleaned with a 0.05 M HCl solution.

**Figure 6 membranes-12-00290-f006:**
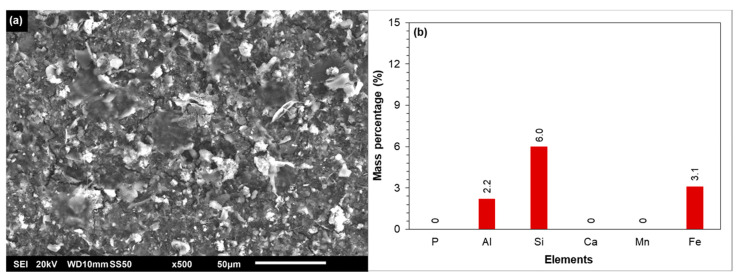
(**a**) SEM image and (**b**) EDX analysis of the 0.45 µm filter after filtration of the 0.05 M HCl solution (of the tail element of the first stage).

**Figure 7 membranes-12-00290-f007:**
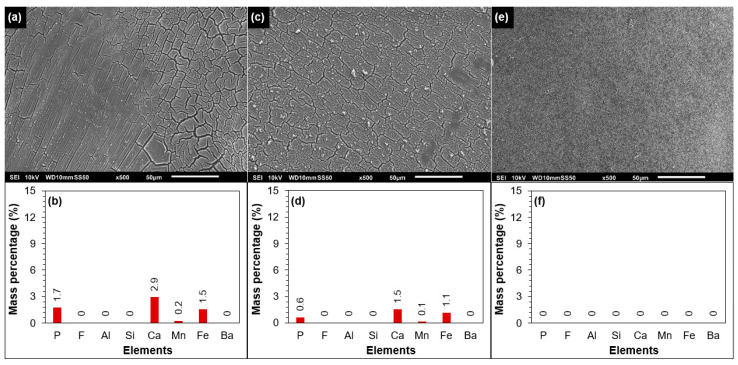
(**a**) SEM and (**b**) EDX analyses of the fouled tail element of the third stage. (**c**) SEM and (**d**) EDX analyses of the tail element of the third stage cleaned with 0.05 M NaOH solution. (**e**) SEM and (**f**) EDX analyses of the tail element of the third stage cleaned with 0.05 M HCl solution.

**Figure 8 membranes-12-00290-f008:**
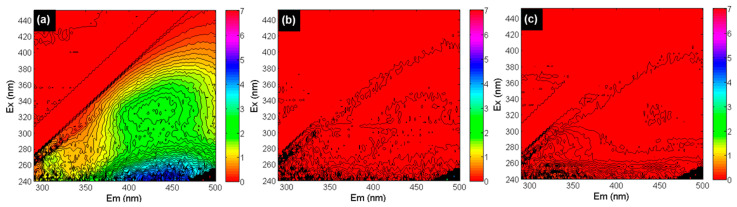
FEEM analysis of the (**a**) RO concentrate at 85% recovery, (**b**) 0.05 M NaOH cleaning solution, and (**c**) 0.05 M HCl cleaning solution (Ex and Em represent excitation and emission wavelengths).

**Figure 9 membranes-12-00290-f009:**
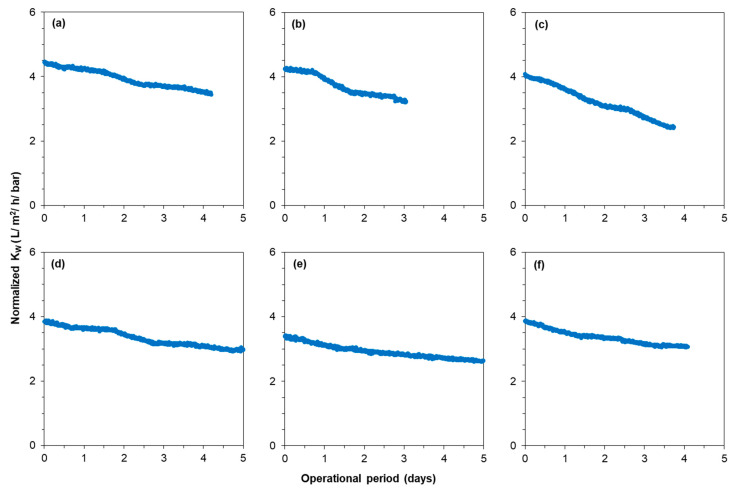
Average normalized permeability of the last stage of the RO unit at 85% recovery (**a**) without antiscalant, (**b**) with 2.5 mg/L AS–1, (**c**) with 2.5 mg/L AS–2, (**d**) with 2.5 mg/L AS–3, (**e**) with 5 mg/L AS–4, and (**f**) with 5 mg/L AS–5.

**Figure 10 membranes-12-00290-f010:**
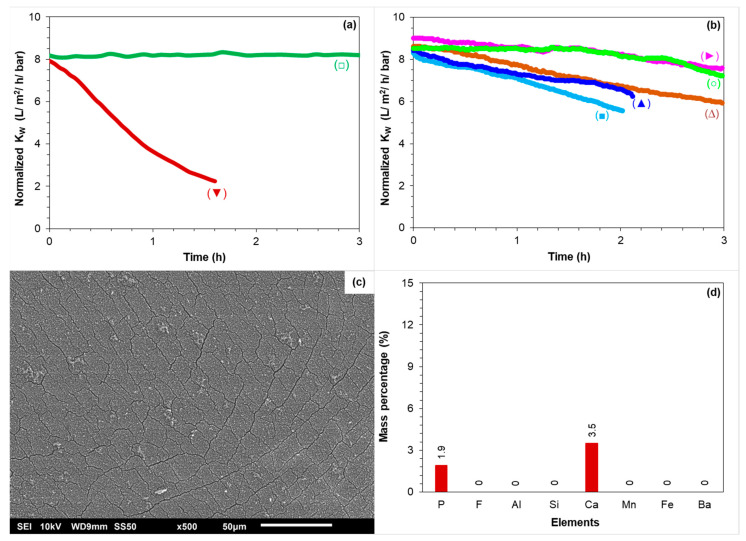
(**a**) Normalized K_w_ of the small RO element when fed with the synthetic concentrates of (□) 80% recovery without antiscalant and (▼) 85% recovery without antiscalant. (**b**) Normalized K_w_ of the small RO element when fed with the synthetic concentrates of 85% recovery with 33.3 mg/L of (■) AS–1, (▲) AS–2, (○) AS–3, (∆) AS–4, and (▶) AS–5. (**c**) SEM image of the membrane surface fouled with calcium phosphate without antiscalant addition. (**d**) EDX analysis of the membrane fouled with calcium phosphate without antiscalant addition.

**Table 1 membranes-12-00290-t001:** Feedwater (anaerobic GW) composition.

Cations	Concentration (mg/L)	Anions	Concentration (mg/L)
Calcium	115.2	Sulphate	43.4
Magnesium	17.4	Chloride	113.6
Sodium	55.2	Fluoride	0.1
Potassium	5.6	Bicarbonate	391.8
Barium	0.1	Carbonate	-
Strontium	0.5	Nitrate	0.2
Iron (II)	8.5	Silica	16.7
Ammonium	3.7	Orthophosphate	2.1
Other properties of the RO feed:			
pH	7.0	TDS (mg/L)	750–800
Temperature (°C)	12	DOC (HS) (mg/L)	8.6 (5.3)
Turbidity (NTU)	<0.1		

**Table 2 membranes-12-00290-t002:** Tested antiscalants as recommended by suppliers for increasing RO recovery to at least 85%.

Antiscalants ^▲^	Chemical Nature	Target Scalants	
		Primary Scalants Targeted	Additional Scalants Targeted
AS–1	Blend of phosphonates and carboxylic acids	Calcium phosphate/carbonate	Silica, iron/clay fouling, etc.
AS–2	Proprietary acrylic polymer with chelate agent	Silica, calcium phosphate	Calcium carbonate, etc.
AS–3	Information not available	Calcium phosphate/carbonate	Silica, clay, metal oxides, etc.
AS–4	A modified polycarboxylate	Calcium phosphate/carbonate	Silica, etc.
AS–5	Sulfonated polycarboxylate	Calcium phosphate	Silica, calcium carbonate, etc.

^▲^ The antiscalants’ real names are replaced with arbitrary names.

**Table 3 membranes-12-00290-t003:** RO operation without antiscalant (to identify scalants that cause permeability decline), and with antiscalants (to increase RO recovery to 85%).

Run	Pressure Vessel Configuration	Recovery (%)	Antiscalant	Antiscalant Dose ^●^ (mg/L)	Run Period (Days)
A	3-2-1 (6 elements)	80	-	0	10
		85	-	0	32
B	6-2-1 (3 elements)	85	-	0	5
			AS–1	2.5	3
			AS–2	2.5	5
			AS–3	2.5	4
			AS–4	5.0	5
			AS–5	5.0	4

^●^ Tested antiscalant doses were the suppliers’ recommended doses.

**Table 4 membranes-12-00290-t004:** Once-through lab-scale RO tests with the synthetic RO concentrates of 80 and 85% recovery in the absence and presence of antiscalants.

Feed Solution	Antiscalant	Antiscalant Dose ^●^ (mg/L)	Ca^2+^ (mg/L)	PO_4_^3−^ (mg/L)	HCO_3_^−^ (mg/L)	pH (−)
Synthetic concentrate of 80% recovery	-	0	575	10.5	200	7.4
Synthetic concentrate of 85% recovery	-	0	767	14	200	7.6
	AS–1	33.3	767	14	200	7.6
	AS–2					
	AS–3					
	AS–4					
	AS–5					

^●^ The antiscalant dose is the concentration of antiscalant in the synthetic RO concentrate.

**Table 5 membranes-12-00290-t005:** Atomic concentrations of various elements (obtained via XPS analysis) of the fouled membrane.

Elements	C	N	O	P	S	Ca	Mn	Fe
Average atomic concentration (%)	46.35	1.9	38.47	3.45	1.18	6.19	0.47	1.98
Standard deviations	0.9	0.29	0.48	0.42	0.06	0.32	0.17	0.26

**Table 6 membranes-12-00290-t006:** The binding energies of various elements on the fouled membrane (determined via XPS analysis).

Element	Binding Energy (eV)	Identified Compound(s) According to the NIST Database
Carbon (C1s)	284.8	Reference value
Calcium (Ca2p_3/2_)	347.2	Ca_3_(PO_4_)_2,_ Ca_8_H_2_(PO_4_)_6_·5H_2_O, Ca_10_(PO_4_)_6_(OH)_2_
Phosphorous (P2p_3/2_)	132.9	Ca_3_(PO_4_)_2_
Iron (Fe2p_3/2_)	711.1	Fe_2_O_3_
Manganese (Mn2p_3/2_)	642.4	MnO_2_

## Data Availability

Not applicable.

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
