# Peer review of "Foulant Identification and Performance Evaluation of Antiscalants in Increasing the Recovery of a Reverse Osmosis System Treating Anaerobic Groundwater"

_membranes, 2022, doi:10.3390/membranes12030290_

Round 1
Reviewer 1 Report
The manuscript titled “Foulant Identification and Performance Evaluation of Antiscalants in Increasing the Recovery of a Reverse Osmosis System Treating Anaerobic Groundwater” and written by M. Nasir Mangal et al. is interesting and it is well written and structured. I recommend a very little minor revision based on the following comments:
- Page 2, lines 63-70. The authors were right by remarking the impact of scaling in RO systems. I only miss a couple of references to support the paragraph. Studies related with the estimation of maximum water flux recovery considering different feedwater and antiscalants and the antiscalant operation costs in full-scale BWRO systems that desalinate groundwater.
- I know that the author did the abbreviation of RO in the abstract but, it should be also done in the manuscript. Page 1, line 39, please, write reverse osmosis (RO). Revise the document considering the rest of abreviations.
Reviewer 2 Report
Hello,
In general, the article is tackling a very important and interesting topic, as it discusses the effect of antiscalants on water recovery during the treatment of groundwater using reverse osmosis membranes. The manuscript is well-written, well-structured, and self-explanatory. However, a few comments summarized as follow have to be considered:
- The authors stress the term “anaerobic groundwater” rather than the general term groundwater, so some explanation for the term used should be provided.
- OMPs and FEEM need to be identified when mentioned for the first time.
- Authors should discuss and specify the strict regulations and standards motivating the increase of recovery from 80 to 85%, and the significance of such increase.
- I do not think that there is a need for subsection 2.1.1 as an independent section, and it can be part of 2.1.
Regards
